# Low-Fouling Substrates for Plasmonic Sensing of Circulating Biomarkers in Biological Fluids

**DOI:** 10.3390/bios10060063

**Published:** 2020-06-10

**Authors:** Elba Mauriz

**Affiliations:** 1Department of Nursing and Physiotherapy, Universidad de León, Campus de Vegazana, s/n, 24071 León, Spain; elba.mauriz@unileon.es; 2Institute of Food Science and Technology (ICTAL), La Serna 58, 24007 León, Spain

**Keywords:** nanoplasmonics, low-fouling, circulating biomarkers, biological fluids, SPR, LSPR, SERS

## Abstract

The monitoring of biomarkers in body fluids provides valuable prognostic information regarding disease onset and progression. Most biosensing approaches use noninvasive screening tools and are conducted in order to improve early clinical diagnosis. However, biofouling of the sensing surface may disturb the quantification of circulating biomarkers in complex biological fluids. Thus, there is a great need for antifouling interfaces to be designed in order to reduce nonspecific adsorption and prevent inactivation of biological receptors and loss of sensitivity. To address these limitations and enable their application in clinical practice, a variety of plasmonic platforms have been recently developed for biomarker analysis in easily accessible biological fluids. This review presents an overview of the latest advances in the design of antifouling strategies for the detection of clinically relevant biomarkers on the basis of the characteristics of biological samples. The impact of nanoplasmonic biosensors as point-of-care devices has been examined for a wide range of biomarkers associated with cancer, inflammatory, infectious and neurodegenerative diseases. Clinical applications in readily obtainable biofluids such as blood, saliva, urine, tears and cerebrospinal and synovial fluids, covering almost the whole range of plasmonic applications, from surface plasmon resonance (SPR) to surface-enhanced Raman scattering (SERS), are also discussed.

## 1. Introduction

The application of personalized medicine in clinical practice requires the repeated assessment of noninvasive biomarkers [1]. The identification and quantification of biomarkers in biological samples is of utmost importance for diagnostic, prognostic and therapeutic purposes [2]. Clinical validation of biomarkers is also needed for predicting the risk of developing a disease, characterizing the biological response to treatment and monitoring the extent of adverse effects [3]. Most indicators of disease progression and therapeutic interventions are circulating molecules ranging from low-molecular-weight compounds (e.g., salivary cortisol, blood glucose, genetic markers) to larger compounds such as proteins and tumor cells [4]. Biomarker analysis has been commonly performed in localized biological tissues because of its ease of detection at higher concentrations. However, acquisition of tissue samples is difficult, provides heterogeneous information and is only indicated when the disease has progressed. As an alternative, monitoring of biomarkers in biofluids presents remarkable benefits over tissue biopsies since it is easier to obtain through minimally invasive procedures. The quantification of circulating biomarkers also provides real-time information from the early stages of numerous diseases, including cancer, stress, brain damage, mental and endocrine disorders, infections or heart disease [3,5,6,7,8].

Obtaining circulating biomarkers from easily acquired samples offers the possibility of serial monitoring during routine physical examinations. Common clinical applications comprising monitoring biomarkers in blood, saliva, urine, cerebrospinal fluid, synovial fluid, tears, sweat, vaginal secretion and ascites fluid [9] have been already reported by conventional methods such as polymerase chain reaction (PCR), enzyme linked immunosorbent assay (ELISA), radioimmunoassay (RIA), immunohistochemistry (IHC), nanoparticle tracking analysis (NTA), dynamic light scattering (DLS), next-generation sequencing (NGS) and flow cytometry [10,11,12]. Nevertheless, continuous read-out analysis through reliable noninvasive techniques still remains a challenge.

From this perspective, biosensing technologies and, more specifically, plasmonic platforms provide a solid framework towards the development of point-of-care devices for clinical diagnostics [13]. On one hand, the utilization of novel configuration approaches in combination with alternative sensing formats enable the development of ultrasensitive plasmonic applications. Likewise, the emergence of chemical functionalization strategies based on the fabrication of nanomaterials and the design of nanopatterned structures impels the integration of biological receptors into multiplexed and miniaturized sensing schemes. Plasmonic optical biosensors benefit from the propagation of surface plasmons confined to the interface between a thin metal layer and a dielectric, being capable of monitoring molecular interactions into resonant spectral shifts due to the changes in the refractive index (see Figure 1) [14].

Recent improvements in analytical performance have enabled the utilization of more sensitive detection formats adapted to multiplexed configurations. Among novel plasmonic strategies based on the well-known principles of surface plasmon resonance (SPR) (Figure 1), localized surface plasmon resonance (LSPR), SPR imaging (SPRi) and surface enhanced Raman scattering (SERS) have already become consolidated alternatives to laboratory-based immunoanalytical and sequencing techniques.

To support SPR-based configurations, plasmonic nanostructures play a crucial role due to their optical and electronic properties. Nanoplasmonic structures provide signal enhancement and higher spatial resolution by optimizing the assembly of optical components in various detection methods [16,17]. Specifically, the design of nanostructures that control the size, shape and composition of these optical components leads to the precise confinement of the electromagnetic field in spots of only a few nanometers [18]. The utilization of plasmonic nanostructures also allows the enhancement of the electromagnetic field, improving the surface-to-volume ratio and the sensitivity of the systems.

Many comprehensive reviews have thoroughly reported the physical principles of SPR [16,17,19,20,21,22,23,24]; as such, only a brief discussion will be included here. In short, the differences between SPR and LSPR are based on LSPR’s utilization of zero-dimensional metallic nanoparticles instead of two-dimensional thin metallic layers [22]. When the electronic field of metallic nanoparticles is below the wavelength of incident light, free electrons oscillate, collectively generating the phenomenon of LSPR (Figure 2). As with SPR, changes in the refractive index of the surrounding media induce a shift in the LSPR peak position that depends on the geometry of the metallic nanoparticle. Since the sensing area is less limited when using metallic nanoparticles, LSPR may provide better opportunities for the fabrication of multiplexed and miniaturized platforms.

The local electromagnetic field can be enhanced in certain surface spectroscopies, such as Raman scattering [16]. SERS sensors benefit from the longer decay of the electromagnetic field of surface plasmon polaritons (SPP) in comparison with LSPR. Hence, SPP can be modulated at a greater distance from the metallic nanoparticle surface, allowing molecules in the vicinity to experience the electromagnetic field and the subsequent enhancement of Raman scattering [25]. As a result, the sensitivity of SERS is higher than that of SPR or LSPR, thus providing a strong basis for single molecule detection.

Other SPR detection formats, such as SPRi, take advantage of two-dimensional array sensing to simultaneously detect thousands of molecular interactions on the same sensor chip [21,22,23]. In this way, the intensity of the reflected light collected in a CCD camera depends on the refractive changes occurring in each spot of the sensor chip. One of the main advantages of SPR imaging over other SPR-based arrangements is the possibility of automated analysis by maintaining the spatial resolution of the different sensing areas. Finally, SPR systems can exploit the propagation of electromagnetic radiation using optical fibers to transmit radiation from a light source to a detector [23]. To reach total internal reflectance, fibers with polymer or metal cladding are coated with a thin gold layer and a buffer layer between the fiber and the metal film. SPR fiber-optics approaches have been mainly used to improve the miniaturization capacity of plasmonic applications.

Therefore, current plasmonics applications are taking advantage of the functionalization of nanomaterials to improve the detection of a broad range of biomarkers, avoiding time-consuming multistep procedures and highly demanding biofluid requirements [18]. Particularly, novel plasmonic approaches based on the chemical activation of different types of nanoparticles have been applied to the determination of a broad range of biomarkers from circulating tumor-associated nucleic acids to living cells, proteins or exosomes [7,26,27,28]. In the same vein, the development of nanostructures also relies on the fabrication of plasmonic surfaces based on optical antenna structures such as arrays of nanoholes, nanoslits, nanostars or nanodiscs [29,30,31]. The design of ordered arrays of nanoparticles by lithographic patterning has permitted the improvement of plasmonic substrates, providing high-spatial-resolution surface structures (Figure 3). Although the most common lithographic techniques are performed before biochemical functionalization, the fabrication of plasmonic nanoantennas has allowed single cell detection in complex media [16,17,18,32,33]. Additionally, the combined used of plasmonic metals and nanopores inserted in living cells opens a promising way towards the development of a new generation of nanoplasmonic tools.

Although the versatility of nanoplasmonic biosensing has permitted label-free detection of multiple biomarkers in biofluids at low detection levels, many plasmonic applications are still limited by nonspecific adhesion of proteins and cell matrices. To overcome this problem, antifouling materials have been developed in order to prevent the interference of undesired molecular interactions [34,35,36,37]. Therefore, the design of antifouling coatings is of paramount importance for preserving the long-term stability of bioactive nanosurfaces in complex biological media. In this context, extensive efforts have been directed towards the fabrication of high-throughput nanoplasmonic systems that can be fully integrated as point-of-care devices in decentralized clinical settings [10,38,39].

This article reviews new trends in the analytical performance of nanoplasmonic biosensors for the determination of circulating biomarkers in blood, urine and saliva samples. Special attention has been focused on novel antifouling approaches that succeed in the detection of cancer, inflammatory, infectious and brain-damage biomarkers using noninvasive biosensing detection formats.

## 2. Antifouling Surfaces

As mentioned above, the improvement of plasmonics applications has been primarily focused on achieving ultrasensitive detection limits. For this purpose, several strategies have made use of the amplification of transducer signals by designing optimal substrates based on nanostructures of higher refractive index sensitivity [11]. However, nonspecific binding results in low signal-to-noise ratios and lower sensitivity analysis [34,37]. Specifically, the adhesion of nonspecific matter to the bioactive surface of sensor chips may generate large noise signals that disrupt the biosensing response, hindering the read-out triggered by the target analyte [40]. Therefore, further attempts need to be concentrated on the development of biointerfaces capable of preserving the activity of the biorecognition element immobilized on the nanostructured surface while maintaining unaltered the sensitivity and signal response.

In this sense, several plasmonic-based platforms have utilized low-fouling surfaces to prevent nonspecific adsorption of proteins, cells, lipids and microorganisms from biological samples such as whole blood (plasma or serum), saliva or cell lysates without inhibiting the recognition of analytes and the attachment to biomolecular receptors [34,41]. Since the analytical performance of plasmonic biosensors enables biomarker detection in the nanomolar to picomolar range, antifouling coatings should confer excellent stability and compatibility to the immobilized biological receptor in order to allow analyte recognition at very low concentrations with the sufficient accuracy and sensitivity. Hence, the design of antifouling materials should rely on the formation of hydration layers and/or the utilization of long-chain polymers where steric hindrance may contribute to both suppress undesired fouling and resist nonspecific adsorption [11,34,40].

A variety of antifouling coatings, including alkane thiolate self-assembled monolayers (SAMs), poly(ethylene glycol)/oligo(ethylene glycol) (PEG/OEG)-based materials, zwitterionic compounds (such as phosphorylcholine-based derivatives and betaines), polysaccharides, peptides, mixed-charge polymers, hydrogels or DNA tetrahedron probes, have been successfully implemented using plasmonic biosensors (Figure 4) [34,40,42]. Additionally, traditional antifouling blocking agents such as bovine serum albumin (BSA) or Tween 20 surfactant could be added to the same sample or in the running buffer to reduce nonspecific adsorption.

### 2.1. SAMs and Hydrophilic PEG/OEG Polymers

Among molecular structures with antifouling properties, short-chain alkanethiolated SAMs provide ordered architectures that generate hydration layers. The formation of hydrogen bonds after exposure to bulk water improves the resistance and prevent the adsorption of proteins and other fouling matter [34]. The hydrophilicity of hydroxyl-terminated SAMs and the formation of secondary helical conformations also contribute to reduce protein adhesion due to either the stability of the hydrogen bonds or the density of the packed structure [43]. Since the hydration layers are affected by the electrostatic interactions between polymers and proteins, the distribution of charge density in SAMs with opposing charges presents better resistance to protein adhesion.

Owing to the versatility of SAMs, a wide range of terminal functional groups could be used as bifunctional molecules for the biosensing of biomarkers in clinical analysis [30]. For instance, the functionalization of SAMs with PEG/OEG-based materials prevents protein adhesion due to the steric hindrance and hydrophilicity generated by the long-chain polymers. In brief, when undesired biomolecules coming from the biofluids approach the PEG/OEG surfaces, the combined effect of entropy losses caused by steric hindrance and the hydration layers contributes to minimize protein adsorption [44]. The electrostatic interactions resulting from van der Waals forces hinder adhesion of other water-soluble molecules because of the low compression tendency of PEG polymers [18]. PEG and OEG layers also improve the biocompatibility of plasmonic biosensing chips by leaving the active functional groups easily available for the immobilization of the biomolecular receptor. Since the preparation of active PEG/OEG sensing areas is relatively simple, a broad spectrum of grafting strategies have been successfully applied as antifouling strategies. Although most PEG/OEG layers are covalently coupled with alkanethiol monolayers or cross-linked with a peptide to the inert area, recent approaches have taken advantage of poly(ethylene oxide) layers end-tethered to polydopamines [18,45] to improve the stability and antifouling properties of the surface.

Despite these advantages, PEG and OEG coatings can be easily oxidized in the presence of oxidants and require molecular weights higher than 2000 Da to keep the colloidal stability of the nanostructured substrates [34]. To solve these limitations, PEG/OEG chains may be chemically modified by conformational changes in the chain configuration and density. In this sense, the utilization of a chemical vaporization process may aid in minimizing the adsorption of proteins and other fouling compounds.

### 2.2. Zwitterionic Compounds

Other surface chemistries benefit from the hydrophilicity and charge balance of zwitterionic compounds. Zwitterionic interfaces are bioinert, compact ordered layers comprised of positively and negatively charged groups that suppress nonspecific adsorption of approaching proteins through nonfouling hydration layers. The antifouling properties of zwitterionic coatings respond to the electrostatic interactions provided by the superhydrophilicity of polypeptides, betaines and phosphorylcholine polymers. Hence, the study of zwitterionic electrolyte layers at different pH values has shown that protein adsorption is mostly inhibited when the layer is electrically neutral [46]. Similarly, the formation of hydration layers as a result of hybridization with water by ionic solvation has proved to be effective in producing low-fouling surfaces with minimal protein adhesion (0.2 ng cm^−2^) [47].

Moreover, the capability of zwitterionic compounds to resist fouling adsorption can be maintained after the functionalization of biomolecular receptors. Therefore, the complementary effect of antifouling resistance and specific binding on the performance of zwitterionic layers has been described in several studies [48,49]. For example, thiol-terminated monolayers functionalized with methacrylic acid and phosphorylcholine or betaine derivative polymers have been used to specifically detect the target analyte at high sensitivity levels while protecting the sensing area from foulant compounds [50]. Likewise, the thickness of zwitterionic functional layers (25–45 nm) is closely related to protein adsorption [34]. An interesting approach for evaluating the influence of chain densities involves the design of peptoids with equimolar analogs of opposing signs functionalized with thiol groups and grafted onto Au surfaces (~1 pg/mm^2^) [51]. The application of 3D-structured zwitterionic carboxybetaine thin film hydrogels onto plasmonic sensor chips has also proved its effectiveness for preventing protein fouling or bacterial infections (<5 ng mL^−1^ of foulants in undiluted serum) [41,52,53].

### 2.3. Polysaccharide-Based Materials

Polysaccharides are well-known low-fouling materials due to the ability of their hydrophilic groups, primarily carboxyl and hydroxyl, to form hydration bonds with interacting water molecules. Dextran-based hydrogels are some of the most popular polysaccharide layers with antifouling properties. Dextran coatings have been used for decades in common immobilization procedures to functionalize SPR sensor chips in a wide range of commercial applications [54]. A more recent approach involves the utilization of hyaluronic acid covalently attached to sensor chips. The functionalization with hyaluronic acid provides inert and stable ultralow-fouling surfaces (3 ng cm^−2^) thanks to the hydration layers formed by the amide and carboxyl groups of the disaccharide unit [41,55]. Several surface chemistries consisting of hyaluronic-acid-modified sensor chips have been proven to reduce protein adsorption (0.6–16.1 ng cm^−2^) in complex media when using hyaluronic acid in combination with alkanethiol or polydopamine films [36,56]. However, the antifouling capabilities of polysaccharide-based layers in both single-protein and complex systems are lower than those of PEG or zwitterionic coatings, and their use still remains focused on the functionalization of sensing surfaces.

### 2.4. Other Antifouling Materials

Despite the promising antifouling properties of PEGs, zwitterionic compounds and polysaccharides, it is difficult to obtain materials that satisfy the ideal requirements for the fabrication of stable, biocompatible, low-cost nonfouling substrates [40].

Among novel strategies, DNA tetrahedron probes have the advantage of their structural conformation that provides steric hindrance and hydrophilic environments when measuring microRNA and other clinical biomarkers in crude biological samples [41]. The rigidity of DNA tetrahedron structures also enables the stability of the immobilized biomolecular receptor while detecting the target analyte in complex matrices without adsorption of foulant matter. However, further studies are needed to exploit the antifouling properties of DNA tetrahedrons in preventing nonspecific binding.

Furthermore, other materials such as polyacrylamide, polydopamine and glycoproteins have been described as fouling-resistant compounds capable of reducing protein adsorption and bacterial and cell adhesion. Although additional studies are needed, these materials have exhibited protein adsorption levels comparable to PEG and zwitterionic coatings.

## 3. Plasmonics Applications

Clinical applications in the nanoplasmonics field cover all aspects of optical technologies, from well-established SPR to consolidated strategies involving LSPR, SPRi and more recent advancements in SERS. Nonetheless, the development of analytical improvements for the determination of biomarkers in clinical settings mainly relies upon the complexity of the biological media. The latest advances in clinical diagnostics by plasmonic biosensors can be further classified on the basis of the characteristics of biological samples (see Table 1).

### 3.1. Blood

Common clinical diagnosis utilizes blood, in the form of plasma or serum, as the most exploited biofluid for the determination of biomarkers. Owing to its function and physical properties, a large number of biomarkers can be identified in blood samples. Standard blood panels test biochemical parameters (glucose, lipids, enzymes, electrolytes, creatinine and urea) and immuno-hematological parameters (hemoglobin, red and white cell counts and platelet count) [4]. Other biomarkers associated with disease progression or response to therapy, including nucleic acids, exosomes, circulating cells and other related compounds, have also been the subjects of routine blood analysis.

Since blood samples are easily acquired and provide information allowing the understanding of the comprehensive pathophysiology of many medical conditions, the latest research has been focused on the rapid detection of circulating biomarkers in blood serum. However, blood (and more specifically serum) is a very complicated matrix with a protein content of 60–80 g L^−1^ and a dynamic range that can potentially span 10 orders of magnitude, thereby increasing the possibility of nonspecific binding to either the sensor surface or the analyte [4,9,34].

In this context, plasmonic biosensing offers a new perspective for monitoring blood parameters via noninvasive methods, although challenges derived from nonspecific protein adsorption need to be addressed to enable high-throughput applications. With this purpose, recent advancements in the development of plasmonic-based applications for direct detection of biomarkers in blood samples are classified according to the nonfouling strategy and configuration of the optical device. Limitations concerning the interference of the matrix media, such as protein adsorption and cell adhesion, are also considered.

#### 3.1.1. Nucleic Acids

Circulating nucleic acids (cNAs) are genetic biomarkers involved in the pathophysiology of many diseases. Since cNAs are released to the bloodstream from cells [56], they can be monitored in blood samples without the need for invasive procedures such as tissue biopsies. Although nucleic acids circulating in blood are commonly associated with cancer diagnosis, cNAs are also implicated in the regulation of gene expression of several disease processes, including brain damage and neuronal injury [6].

Among the main categories of cNAs, circulating tumor DNA (ctDNA) and microRNAs (miRNAs) have been receiving increased attention for use in biosensor-based detection strategies. The number of plasmonic applications dedicated to the monitoring of cNAs in blood samples is growing rapidly, although the size of ctDNA fragments and miRNA molecules has limited clinical analysis primarily to LSPR and SERS detection schemes [57].

##### MicroRNA

Mature miRNA are short strands of noncoding RNA that play a pivotal role in the regulation of gene activity and cell processes. The dysregulation of miRNA is associated with the proliferation and metastasis in cancer progression, along with the pathogenesis of diabetes, neurodevelopmental disorders, cardiovascular disease or hematological disorders. Since miRNA expression in peripheral circulation can be connected with disease outcomes, the monitoring of miRNAs in bodily fluids represents a promising approach in clinical diagnosis and drug therapy [5,57,58]. While miRNAs have already been exploited as circulating biomarkers by conventional sequencing methods such as PCR, recent plasmonic configurations may also be compatible with the analysis of low concentrations of miRNAs in complex matrices.

One singular approach involving the determination of miRNA in 100% human serum and cancer cell lysates takes advantage of an antifouling surface consisting of DNA tetrahedron probes (DTPs) covalently attached to gold SPR sensor chips [40]. The formation of DTPs-Au film yielded ultralow adsorption amounts (≤8.0 ng/cm^2^) using two proteins (myoglobin and human albumin serum) in five complex matrices (100% serum, 100% plasma, 9.85 × 10^8^ red blood cells/mL, 5% whole blood and cell lysate). Additionally, the sensitivity of miRNA detection was enhanced by integrating DTPs-Au film with the catalytic growth of AuNPs. The target let-7a was detected at femtomolar levels (0.8 fM) and selectively differentiated from a homologous family. This method shows an original low-fouling alternative to conventional antifouling strategies that enables the sensitive and specific detection of miRNAs in undiluted human serum while maintaining nonspecific binding of interfering proteins (Figure 5).

Another interesting low-fouling approach employing carboxy-functional brushes of zwitterionic homo- and copolymers is presented by Lísalová et al. for examining the capacity of the biological receptor to resist biofouling from blood plasma [59]. The analytical performance of the zwitterionic-activated surfaces was also demonstrated via the sensitive detection of miRNA (miR-16) at 6 nM using an SPR biosensor. Nevertheless, the investigation is primarily focused on the study of the physicochemical processes involved in the functionalization of carboxy-functional groups by comparing copolymer brushes containing zwitterionic carboxybetaine side chains with conventional carboxy-terminated OEG-SAMs. The mechanism of activation of carboxy groups demonstrated that the fouling resistance was only recovered by covalent attachment of amino acid deactivation agents. Hence, low-fouling properties of the zwitterionic layer were improved by further combination with random copolymer carboxybetaine methacrylamide and N-(2-hydroxypropyl) methacrylamide and showed better fouling resistance than homopolymer pCB brushes and OEG-SAMs.

Highly sensitive detection of miRNA15a in spiked human serum samples has also been described using SPR imaging [60]. This research concentrates on the development of an orthogonal signal amplification strategy based on the in-plane and vertical addition of mass. The effective 3D use of the sample spot was achieved by the inline addition of mass through the coupling of miRNA-initiated DNA–DNA hybridization and the vertical addition of mass obtained via the elongation of a DNA-initiated upward polymerization reaction. By combining the signal amplification approach with the design of an SPRi sensing chip comprising isolated gold islands bordered by hydrophobic CYTOP fluoropolymer, a 107-fold improvement of the sensitivity (LOD = 0.56 fM and LOQ = 5 fM) was observed when measuring miRNA15a in 10% diluted human serum. Although worse sensitivity values were obtained by comparing 40% diluted cancer patient (3.46 fM) and healthy (10.5 fM) serum, the capacity of the proposed method to prevent nonspecific binding was demonstrated by recoveries ranging from 98.6% to 104.9% in spiked human serum samples.

The development of a surface-enhanced Raman scattering (SERS) biosensor for the detection of myocardial-infarction-related miRNA (miR-133a) has also been reported [61]. Although the application is focused primarily on the design of Ag/Au nanosphere-based SERS probes coupled with target-catalyzed hairpin assembly (CHA), SERS performance was evaluated by comparing the SERS spectrum obtained from buffer and 10% human serum in the 1 fM to 10 nM range with a limit of detection (LOD) of 0.306 fM. The nonbinding capacity of the surface was explained by the stability displayed through the Ag/Au bimetallic SERS probes. However, further interference of nonspecific protein adsorption has not been explained in detail.

##### Circulating Tumor DNA

Circulating tumor DNA (ctDNA) is extracellular DNA comprised of single- or double-stranded DNA that can be found in numerous biological fluids from blood to synovial or cerebrospinal fluid. Monitoring of ctDNA in the peripheral blood may be a very useful way to obtain information about potential gene mutations. Because ctDNA fragments with more than 150 base pairs could limit plasmonic detection in nanoparticle-based approaches [57], only a recent strategy takes advantage of an LSPR gold-nanorod-based platform for the sequence-specific detection of ctDNA point mutations in spiked healthy patient serum. To investigate specific point mutations, peptide nucleic acid (PNA) probes were conjugated to gold nanorods, and the LSPR absorbance was measured after exposure to synthetic ctDNA at various concentrations [62]. A linear working range below 125 ng mL^−1^ ctDNA and an effective limit of detection of 2 ng mL^−1^ in solution were found when comparing Tris-EDTA buffer and serum measurements. However, the interference of the serum matrix yielded poor specificity results since the discrimination between the mutant ctDNA and the wild type ctDNA was not satisfactorily observed.

##### Circulating Tumor Cells

Monitoring of circulating tumor cells in the circulatory system may be considered a feasible tool for the analysis of fluctuations and treatment response associated with cancer development and dissemination. Nevertheless, the detection of CTCs still remains a challenge since the concentration in blood is very low (1–10 cells/mL of blood) [11,63,64]. Therefore, to enhance their value as clinical biomarkers, robust detection systems that demonstrate sufficient sensitivity are needed.

In this sense, only a few applications have reported CTC monitoring in blood samples.

For instance, CTCs have been detected through a microarray based on grating-coupled surface plasmon resonance (GCSPR) and grating-coupled surface plasmon coupled fluorescence (GCSPCF) imaging [65]. The analysis comprised the immobilization of specific antibodies to cell surface markers that can capture both breast cancer cells and mouse tumor cell lines in blood. To prepare the microchip, various antibodies to tumor cell, leukocyte and plasma protein antigens were pin-spotted on 1 cm^2^ surfaces in triplicate. The printed microchips were assembled and placed in the GCSPR device and blocked with BSA 2% to minimize the interference of the diluted mouse blood (1:100). The association of CTCs to the immobilized antibodies was measured by the SPR angle shift due to the changes in the refractive index. Results are only related to the identification of cell expression of heat shock proteins, while nonspecific binding is only explained because of the multiple components of whole blood samples.

CTCs have been also detected in blood by means of a plasmon-enhanced electrochemistry device [66]. The enhancement of sensitivity is attributed to the combination of LSPR excitation and the electrochemical current response obtained from the plasmonic gold nanostar functionalization of the glassy carbon electrode. On the other hand, the detection of CTCs is based on the molecular recognition provided by the aptamer immobilized on the functionalized surface. The capture of CTCs was quantified in 5 to 10 cells mL^−1^ depending on the cell line.

#### 3.1.2. Exosomes

Exosomes are extracellular nanovesicles that are 30–150 nm in diameter and carry valuable molecular and genetic information contained in proteins, lipids, nucleic acids and chemical messengers implicated in the signaling cascades of tumor progression [11,67]. The feasibility of exosomes as promising clinical biomarkers relies on their stability, resistance to degradation and higher concentration in body fluids [67] once released from tumor cells in comparison to CTCs and ctDNA. Hence, the development of valid detection techniques is mostly focused on the isolation and quantification of exosomes.

A recent approach describes multiplex characterizations of non-small-cell lung cancer derived exosomes by bioaffinity interactions of antibodies using SPRi biosensing [68]. The signal amplification enabled a limit of detection of 104 particles/μL with the aid of functionalized gold nanoparticles. The screening of exosomes was achieved in purified clinical plasma samples. Microarray chips functionalized with antibody-modified Au nanoparticles were exposed to exosomes extracted from plasma of lung cancer patients. The only treatment for minimizing the fouling effect of the plasma matrix was the incubation in 1% (*w*/*v*) BSA and the purification of plasma content by a differential centrifugation protocol.

Similarly, another SPRi bioassay has been applied to the direct detection of exosome subpopulations derived from neurons and oligodendrocytes in blood [69]. Nonspecific binding was prevented by the isolation of exosomes through the use size-exclusion chromatography prior to SPRi analysis. Additionally, several surface chemistries involving mixed layers of PEG molecules were tested to cover the array of modified antibodies. The density of the carboxylated PEGs increased the signal response from 5% to 20%, while the length negatively affected the assay performance. Measuring in the optimized conditions allowed the identification of circulating subpopulations of brain-derived exosomes as well as the quantification of the amount of membrane components of exosomes in each subpopulation. These results open the way to the characterization of exosomes as potential clinical analysis in neurodegenerative diseases.

#### 3.1.3. Cancer-Related Proteins

Cancer-related proteins are a heterogeneous group of biomarkers that provide relevant information concerning the diagnosis, prognosis, progression and response to treatment of the tumor disease. Although cancer biomarker proteins can be overexpressed at elevated concentrations in tumor cells, circulating proteins are present at ultralow concentrations ranging from nanograms to picograms per milliliter in blood once they enter the bloodstream by a diffusion process [11]. However, because of their size, around tens to hundreds of nanometers, circulating cancer proteins can be quantified by specific interaction with antibodies or aptamers through immunoassays and affinity-based techniques [10]. In this context, nanoplasmonic biosensing offers many advantages for the identification and characterization of protein interactions since it enables real-time monitoring of recognition events between target proteins and antibodies or aptamers used as biological receptors.

##### Prostate-Specific Antigen

Despite the huge number of proteins that can be expressed in human cancers, only a small amount are routinely measured in clinical analysis. Among them, prostate-specific antigen (PSA) is one of the most remarkable biomarkers for the determination of prostate cancer. PSA levels indicating high possibility of cancer are commonly measured in serum as total PSA, including free and complexed PSA forms, in the 4.0 to 10.0 ng/mL range. Although there are many analytical techniques involving electrochemical or piezoelectric approaches, several plasmonic applications have been recently applied to PSA detection in serum samples [70].

Kim et al. exploited a fiber-optic localized surface plasmon resonance biosensor using two distinct strategies comprising the integration of a microfluidic channel and the fabrication of nanopatterned gold particles [71]. To demonstrate the feasibility of the proposed method for the early screening of prostate cancer, PSA detection was performed using patient serum samples diluted with deionized water to 13 concentrations. In spite of the lack of antifouling strategies, PSA measurements showed CV values of 13.55% on average for the three concentrations. The comparison with immunoradiometric analysis indicated the underestimation of results at lower concentrations. Nevertheless, a limit of detection of 0.1 pg mL^−1^ was reported using the nanopatterned enhancement approach.

Another singular SPR immunosensor application was developed that enabled the detection of PSA using a mixed system based on the optimization of the pH values of the carrier fluid [72]. PSA was detected in diluted serum samples by tuning the pH of the solution mixture comprising BSA, PSA and CRP. The system makes use of the antigen molecules as charged particles to measure the variation of the antigen surface charges depending on the isoelectric point and pH of the buffer solution. Serum samples were first diluted serially to 10, 100, 1000 and 10,000 times by adding increasing quantities of PBS buffer at pH 6.5, and they were subsequently spiked with PSA solutions at a concentration of 0.01 mg mL^−1^. The amounts of interfering proteins were ~66 times that of the amount of PSA for samples diluted 100 times or more. Although analytical parameters concerning the assay performance are not reported, this work proves the influence of varying the pH of the solution to detect the amount of capture.

##### Carcinoembryonic Antigen

Carcinoembryonic antigen (CEA) is a cell adhesion glycoprotein broadly used as cancer biomarker. CEA levels in serum are elevated above 5.0 ng mL^−1^ in many malignancies, thus indicating possible cancer in a variety of locations, including gastrointestinal, breast, liver, pancreatic and lung cancers. Therefore, sensitive detection of CEA in serum is needed for cancer diagnostics and treatment.

Several novel functionalization strategies have been described to avoid nonspecific adhesion of interfering proteins by plasmonic biosensing while enabling CEA detection with high sensitivity. An SPR biosensor involving the biological functionalization of nanoparticles using varying concentrations of either streptavidin or anti-CEA specific antibody was tested to measure nonspecific sensor responses, reproducibility and colloidal stability [73]. Since the study was concentrated on predicting the enhancement of the SPR in both PBS and blood plasma by using the z-potential, it did not include analytical parameters related to assay sensitivity. In contrast, the specificity of SPR responses were measured simulating in vivo conditions by immersing biofunctionalized nanoparticles in blood plasma for several hours. The comparison between streptavidin and antibody biofunctionalized nanoparticles revealed similar results regarding specificity, while the reproducibility and nonspecific response yielded good results under complex media conditions.

Another SPR biosensor consisting of the fabrication of metal carbide nanosheets of Ti_3_C_2_-MXene combined with multiwalled carbon nanotube (MWCNT)–polydopamine (PDA)–Ag nanoparticles (AgNPs) as signal enhancers was developed to detect CEA in buffer and patient serum samples [74]. By making use of the hydrophilic, biocompatible surface provided by Ti_3_C_2_-MXene and the capacity of the Ti_3_C_2_-MXene/AuNPs composites to orientate monoclonal anti-CEA antibodies, a sandwich assay format involving nanohybrids conjugated with polyclonal anti-CEA antibodies achieved a limit of detection of 0.07 fM (Figure 6). Other analytical parameters concerning the accuracy and selectivity of results were first proved in buffer and then tested in a complex matrix. For instance, the reproducibility of the sensing platform was evaluated using CEA-spiked healthy human serum samples diluted 10 times in PBS, showing recoveries ranging from 98.2% to 111.1% Finally the applicability for clinical diagnosis was determined by validating five patient serum samples with ELISA (RSD 2.7% to 14.0%).

##### Other Cancer-Related Proteins

Other works associated with the determination of cancer protein biomarkers report similar approaches in either spiked or real diluted serum samples showing good sensitivity results. Several examples involving the detection of alpha fetoprotein hepatocellular cancer biomarker and the CA125/MUC16 ovarian tumor circulating biomarker have employed SERS, SPR and SPRi applications without exploiting any other antifouling strategy besides the dilution of serum samples [75,76,77]. In this line, an SPR-based immunosensor makes use of carboxyl-modified MoS_2_ composites to detect the lung-cancer-associated biomarker cytokeratin 19 fragment (CYFRA21-1) in spiked diluted samples [78]. The fouling effect was measured by evaluating SPR angular shifts for various serum dilutions. To reduce interference and assure minimum cross-reaction, the optimum serum dilution was set to 6.25% (S5) in SRB buffer, showing good specificity and selectivity and a detection limit of 0.05 pg mL^−1^ (Figure 7). The glycan matrix shape provided by the carboxyl-MoS_2_-based SPR chip offers a high surface area that allows the enhancement of the bioaffinity response, suggesting its potential use for clinical diagnostics.

Likewise, the folic acid protein, which is another cancer protein biomarker and is overexpressed in epithelial cancers, has been detected at femtomolar levels in serum by using an SPR biosensor [79]. To prevent nonspecific interactions, graphene-coated sensor chips functionalized with folic acid receptors were immersed in mixtures of human serum and bovine serum albumin (BSA) with a volume ratio of 1:1. The post-adsorption of human serum in combination with BSA resulted in an antifouling interface without affecting the sensitivity and selectivity of the assay. The stability of the sensor surface was also evaluated.

#### 3.1.4. Alzheimer’s Disease Biomarkers

Monitoring of clinically useful biomarkers of Alzheimer’s disease (AD) is crucial for predicting changes in the pathophysiological process from the early stages of the disease. Due to the aging population, the instances of progression of AD-derived neurodegenerative disorders, along with the number of people affected, will rapidly increase in near future [80]. Therefore, the identification of peripheral biomarkers is necessary before the formation of the senile plaques containing overexpressed amyloid-beta protein and the neurofibrillary tangles composed of intraneuronal aggregates of hyperphosphorylated tau proteins become irreversible.

For instance, amyloid beta 1-42 (Aβ42) has been detected in plasma and serum by an SPRi assay through an antibody-mimetic peptoid nanosheet [81]. To reduce nonspecific binding, peptoid nanosheets were immobilized on a chip coated with carboxylated poly(OEGMA-co-HEMA) matrices. The synthesis of peptoid nanosheets along with the modified OEG brushes enabled the effective differentiation between plasma and serum of patients with mild cognitive impairment and healthy controls (Figure 8). The specificity and selectivity of the peptoid nanosheets was evaluated at various dilution ratios, showing diminishing binding signals for the highest dilution ratios. A linear relationship between the binding signals and Aβ42 concentrations was found in the 1 to 10,000 pM range for 1:4000 dilutions.

Regarding the biosensing of tau protein, a multichannel SPR application reports the detection of the Tau-381 isoform at femtomolar levels in undiluted plasma samples through a DNA aptamer/antibody sandwich assay [82]. The covalent coupling of the DNA aptamer was achieved by modifying the surface chemistry of SPR sensor chips via different ratios of a mixed monolayer of either two differently terminated alkanethiols or PEG molecules (60% 11-mercaptoundecanoic (MUA) to 40% 11-mercaptoundecanol (MUD), 40% MUA to 60% MUD or 60% MUA to 40% PEG monolayer surfaces). Blocking of nonspecific interactions was more effective using alkanethiol mixtures due to the uniformity of the packed self-assembled monolayers. Additionally, the comparison with ELISA determinations yielded better assay performances in terms of sensitivity. The improvement in results was attributed to nonspecific adsorption on the SPR chip and the use of continuous flow microfluidics.

#### 3.1.5. Inflammation Biomarkers

Inflammation biomarkers are associated with biological responses at the cellular and molecular levels in chronic diseases. They are also involved in the immunochemical response to infection and oxidative stress. The study of protein signaling pathways is also of interest in identifying the role of inflammation-inducing risk factors in the development of cancer. Although inflammation biomarkers have been traditionally measured by ELISA, the number of plasmonic biosensing approaches has grown in recent years [83,84].

A multiplex SPRi platform was developed for the simultaneous detection of two interleukins (IL-6 and IL-8) and two microRNAs (miRNA-21 and miRNA-155) in 10% serum using a microarray design involving citrate-stabilized Fe_3_O_4_@Au core/shell nanoparticles (NPs) to enhance the signal [28]. The comparison between SPR responses for larger and smaller molecules indicated that signal changes are greater for small oligonucleotide hybridization than for large sandwich protein interactions. The combined effect of the iron oxide/gold core/shell nanomaterial enabled a highly sensitive detection while reducing nonspecific adsorption of foulants present in serum samples. The Fe_3_O_4_@Au NP bioconjugation strategy produced a signal amplification proportional to the serum biomarker concentration of about 3 times for the ILs and 7 times for the miRNAs, thus confirming the selective recognition between the immobilized capture probes and their target biomolecules.

#### 3.1.6. Biomarkers for Infectious Diseases

Infectious diseases are disorders caused by pathogenic agents such as bacteria, viruses, parasites or fungi that can be spread from person to person via direct (e.g., skin contact) or indirect (e.g., via contaminated food or water) contact. Owing to the high capacity for dissemination, it is essential to have sensitive, specific methods for the rapid screening of populations that can respond to treatment in the shortest period of time. Commonly, polymerase chain reaction (PCR), immunoassays and mass spectrometry (MS) tests have been used to detect infection pathogenic microorganisms. Despite their sensitivity, traditional methods are time-consuming and require multiple steps. Therefore, plasmonic biosensing could achieve the point-of-care (POC) perspective recommended by the World Health Organization (WHO) for the control of infectious diseases according to the “ASSURED” criteria: (1) affordable, (2) sensitive, (3) specific, (4) user-friendly, (5) rapid and robust, (6) equipment-free and (7) deliverable to end-users [15].

Hepatitis B (HB) is an infectious disease caused by the hepatitis B virus, with prevalence rates between 10% and 20% in developing countries. The need for performing diagnostic assays outside the laboratory makes the development of plasmonics-based assays very attractive. An SPR biosensor was applied to detect anti-HBs antibodies in serum samples. The assay makes use of a chemically modified surface with antifouling copolymer poly[(N-(2-hydro-xypropyl) methacrylamide)-co-(carboxybetaine methacrylamide)] brush-functionalized with HBsAg [85]. Clinical serum samples diluted at a volumetric ratio of 1:10 were injected in the SPR platform and measured directly through the shift of SPR wavelength. The analytical signal was further amplified by a secondary anti-hIgG antibody. The antifouling effectiveness of the copolymer surface was demonstrated by attaining similar SPR responses and showing the same binding activity for the target analyte and anti-hIgG, thus indicating nonsignificant steric hindrance. This fact was explained by the combined effect of the carboxyl groups and the hydro-xypropyl methacrylamide units for ensuring the biofunctionalization and preserving the antifouling properties of the coated surface, respectively. Additionally, SPR measurements were validated by ELISA method.

Another SPR platform has been developed to detect *Brucella melitensis,* a type of Gram-negative, facultative and intracellular bacteria that causes brucellosis and is considered a potential biological warfare agent [86]. An SPR chip consisting of 4-mercaptobenzoic acid (4-MBA)-modified gold was used to covalently immobilize two different DNA probes of *B. melitensis* and detect complementary DNA fragments. After studying the affinity parameters between DNA targets and the two DNA probes, 10 real samples of *B. melitensis* in various concentrations were analyzed, showing the lowest SPR responses at 1:6400 dilution. Since the linear range is not reported, no other effect associated with nonspecific binding is described.

### 3.2. Saliva

Although body fluid testing is normally performed in blood, it requires vascular access through an invasive procedure involving the injection of a needle in a vein for collection of the sample. Saliva sampling is an alternative to classic biofluid analysis since it provides many advantages over other biofluids for the detection of various clinical biomarkers while ensuring patient comfort. Saliva is mainly composed of water (99%), but it also contains inorganic and organic compounds such as electrolytes, mucus, enzymes, proteins, peptides and lipids [4]. Since saliva is implicated in a variety of physiological processes, from the lubrication of oral tissues to the regulation of homeostasis and bacterial or viral growth, a number of salivary disease-signaling biomarkers can be associated with many systemic disorders. In this sense, thousands of proteins, as well as microRNA (miRNA) transcripts, hormones and other metabolites, are uniformly distributed in saliva and can therefore be measured for monitoring normal and disease conditions. Furthermore, collection of saliva samples is a minimally invasive procedure that can be easily obtained by commercially available oral fluid collectors without causing pain to the patients. The simple storage and transport are also additional benefits in comparison with blood sampling. However, several shortcomings should be overcome, including the heterogeneous content of the saliva matrix and the low levels of salivary biomarkers (in some cases several orders of magnitude lower in comparison to serum samples) [87,88]. Recent developments in plasmonic platforms have demonstrated highly sensitive and selective monitoring of a broad range of biomarkers, including proteomic, genomic and microbiological biomarkers, in saliva samples from early-stage disease detection to progression and treatment response [89,90]. Despite the growing interest of plasmonic sensing concerning the detection of salivary biomarkers, the number of applications is very low in comparison with blood-based determinations.

An LSPR platform has been developed for the direct detection of cortisol in saliva, a steroid hormone associated with stress conditions [91]. The capability to detect cortisol in saliva samples was tested using either aptamers or antibodies as biological receptors. The aptamer-based functionalization strategy involving the immobilization of gold nanoparticles of different sizes yielded more sensitive performances. A limit of detection of 0.01 nM was obtained for the optimized assay conditions. Similarly, the specificity of the assay with respect to structurally related compounds showed significant changes between cortisol and the other tested substances. The assay was ultimately validated by ELISA method, demonstrating good agreement in accuracy between LSPR and ELISA determinations. The efficiency of the proposed method was attributed to the antifouling effect caused by the treatment of salivary samples via filtration.

Another innovative approach takes advantage of angular interrogation of SPR with plasmonically amplified fluorescence detection to monitor hepatitis B antibodies in clinical saliva. Antibodies against hepatitis B surface antigen (anti-HBs) are traditionally determined in serum to confirm infection or the efficiency of vaccination. Detection of anti-HBs in saliva offers a challenging noninvasive strategy that requires higher performance platforms to identify lower antibody concentrations under increased fouling conditions. In this context, the enhancement of the optical signal of a plasmonic biosensor by fluorescence amplification while providing a modified antifouling surface based on brushes of poly[(N-(2-hydroxypropyl) methacrylamide)-co-(carboxybetaine methacrylamide)] has been reported [92]. The antifouling capacity of the betaine brushes was demonstrated by preserving the properties of the biofunctionalized surface with hepatitis B surface antigen. Similarly, the assay sensitivity and accuracy of positive and negative clinical saliva samples was validated with ELISA determinations in serum samples.

An SPR immunosensor was developed for the detection of matrix metalloproteinase-9 (MMP-9), an endopeptidase that requires zinc for its catalytic activity and is closely related with physiological processes concerning inflammation, wound healing, tissue infection and cancer metastasis and progression. For this detection method, an antibody immobilization format was used for measuring metalloproteinase saliva levels in healthy volunteers and patients with periodontal disease [93]. A modified surface consisting of a carboxymethyldextran hydrogel was characterized by atomic force microscopy to optimize the functionalization of anti-metalloproteinase antibodies. Additionally, to enhance the SPR response, the affinity interactions between immobilized antibody and metalloproteinase target analyte were calculated by kinetic parameters involving the equilibrium dissociation constant and the maximum binding capacity. Other analytical parameters such as sensitivity, reproducibility and reusability were also measured, thereby indicating a limit of detection of 8 pg mL^−1^ and recoveries of about 94% in saliva samples. The capacity of the carboxymethyldextran hydrogel to form three-dimensional immobilization matrices allowed the signal amplification in saliva while maintaining the specific detection of metalloproteinase-9 in saliva. However, additional studies on the specificity or the interference of related compounds are needed.

Another singular approach involving the same carboxymethyldextran-modified surface but making use of an SPR imaging biosensor was applied to detect ABH antigens from red blood cell (RBC) membranes in saliva [94]. The aim of the study was to prevent misinterpretation of blood group typing by diminishing the time of conventional agglutination assays based on standard neutralization testing. A multiplex format consisting of an antibody array with immobilized anti-A, anti-B and anti-H on the hydrogel-coated surface was developed to specifically quantify A, B and H substances simultaneously. A sandwich assay with a mixture of anti-A, anti-B and anti-H antibodies was developed to increase sensitivity. The assay demonstrated good specificity and precision when diluting saliva specimens (Figure 9). To avoid nonspecific binding due to the presence of other proteins in the matrix, a 25% (*v*/*v*) saliva concentration should be used. No other antifouling strategy besides a pretreatment of saliva samples that included freezing and boiling was carried out to inactivate salivary enzymes and prevent their adsorption.

### 3.3. Urine

Urine plays an important role regarding the filtration of the blood supply that enters the kidney. The filtrate produced is the result of the excretion and reabsorption processes that occur in the Bowman’s capsule of the nephrons. As a consequence, urine composition resulting from filtration is rich in water, minerals (primarily Na+ and K+), glucose, proteins, metabolites and the waste product, urea [4]. Owing to the qualitative and quantitative data that it can provide about the metabolic state, it is one of most employed biofluids for biomarker detection after blood serum and plasma. Additionally, urine collection is more advantageous since it is available in large quantities via minimally invasive available procedures that allow continuous monitoring and repeated sampling. Further benefits are derived from urine protein concentration, which is about three orders of magnitude less than that of blood and thus results in a less complex matrix with lower levels of potentially interfering factors in routine analysis. Moreover, the urine proteome represents not only urogenital system disorders but also systemic specific diseases involving cancer cell proliferation or endocrine-altered hormonal function. The development of analytical platforms for measuring urinary biomarkers presents a viable alternative to overcome inherent limitations and particular challenges of conventional blood testing [95]. Therefore, most plasmonic applications monitor biomarkers in urine without the aid of nonfouling strategies.

Although glucose determination is normally performed by capillary blood sampling using well-established glucose-oxidase-based assays, a recent approach has sensitively detected glucose in urine samples by a fiber-optic surface plasmon resonance sensor enhanced with phenylboronic-acid-modified Au nanoparticles [96]. The proposed method takes advantage of an Au nanoparticle–glucose–Au film sandwich structure to evaluate the assay sensitivity and selectivity. The lowest concentration of glucose detected was 80 nM, below both physiological blood and urinary glucose concentrations (2.8 mM), while the selectivity was investigated using fructose and galactose as the control experiments. For urinary determinations, urine samples were spiked at different concentrations ranging from 0.05 to 5 mM, although no other analytical parameter related to assay sensitivity or validation was presented to confirm the reliability of the performance.

Another SPR biosensor has been developed to detect human serum albumin, which is the most abundant protein in blood and is associated with many disease conditions involving cardiovascular, hepatic or immunological processes that can lead to kidney damage, in aqueous and urine samples. This method makes use of imprinted nanoparticles comprising ethylene glycol dimethacrylate and N-methacryloyl-l-leucine, functionalized with HSA and attached to SPR sensor chips [97]. The nanoimprinted surfaces were thoroughly characterized to optimize the assay conditions. Likewise, the selectivity was investigated by using hemoglobin and transferrin proteins as competitor molecules. A limit of detection of 1.9 pM was obtained for the optimized assay conditions, whereas the sensitivity and selectivity for urine samples were tested in the 0.15–500 nM range. The reusability and stability of the SPR platform were explained by the robustness provided by the HAS-imprinted sensor chip.

Urinary miRNA biomarker has been measured by a multiplex detection format based on transmission surface plasmon resonance (t-SPR) with the use of capped gold nanoslit integrated microfluidics [98]. To increase sensitivity, the microfluidic chip was combined with functionalized nanoparticles as signal enhancers. The multiplexing ability of the platform was demonstrated by the simultaneous detection of three different biomarkers in spiked urine samples at femtomolar levels. To prevent nonspecific binding, the hybridization time between the functionalized Au film surface and the target molecules was drastically reduced from 1 h to 20 min.

Regarding the monitoring of inflammatory biomarkers, 3-nitrotyrosine (3-NT) has been associated with chronic diseases and is indicative of cardiovascular risk (atherosclerosis), autoimmune disorders (rheumatoid arthritis), neurodegenerative diseases (Alzheimer’s) or endocrine disorders (diabetes). Traditionally, 3-nitrotyrosine is determined by chromatographic methods in serum, urine and tissues. Urine detection of 3-NT has been reported as accomplished by using an SPR immunosensor comprising an indirect competitive assay [99]. The performance of the SPR platform for 3-NT determinations in spiked urine samples yielded better results in terms of sensitivity (LOD 0.12 μg mL^−1^) in comparison with the ELISA method (LOD 0.24 μg mL^−1^). The reliability of the SPR immunosensor was also demonstrated by the recoveries obtained using spiked artificial urine (82.1%–116.6%), which were in accordance with ELISA determinations (86.8%–112.2%).

Finally, a recent SPR application describes the detection of the main immunogenic peptide of wheat gluten in urine samples from healthy volunteers and celiac disease patients [100]. This biosensing approach took advantage of a robust biofunctionalization protocol consisting of the immobilization of prolamin working group (PWG) gliadin on a previously formed PEG/SAM monolayer. An indirect competitive assay conducted with two specific monoclonal antibodies to the epitopes of PWG gliadin peptide enabled sensitive detections in the 1.6 to 4.0 ng mL^−1^ range, depending on the different recognition patterns of the gluten immunogenic peptide. The interference of urine samples on the assay performance was minimized by the combined effect of the ethylene glycol layer and an extra blocking step comprising BSA (10 mg mL^−1^ in PBST) for 2 min before each urine analysis. The addition of BSA enabled the suppression of nonspecific adsorption from the urine matrix without affecting the interaction between the antibody and the PWG gliadin layer. The work also showed good recovery values and correlation with the commercial method for the determination of gluten immunogenic peptides in urine.

### 3.4. Other Biological Fluids

#### 3.4.1. Cerebrospinal Fluid

Cerebrospinal fluid (CSF) is a biological fluid that plays an important role in the distribution of neuroendocrine factors, nutrients and metabolites [4]. Owing to its location, around and inside the brain and spinal cord, CSF can provide relevant information with regard to neurodegenerative diseases, brain damage, inflammation and infection. The content of blood-derived proteins in CSF depends on the entry to the system and varies from ventricular to lumbar collections. Nevertheless, CSF biomarkers are of special interest for the diagnosis of Alzheimer’s disease and other brain-related diseases. In this context, plasmonics represents an emerging trend for the detection of neurodegenerative diseases at early stages.

A representative SPR application consists of surface imprinting of dopamine in a polypyrrole polymer over multiwalled carbon nanotubes (MWCNTs) coated on an optical fiber substrate [101]. To reduce the interference from anionic compounds, a membrane of the cation exchange polymer nafion was deposited over the imprinted sites. The combination of the surface-imprinted carbon nanotubes and the minimal interference conferred by the nafion membrane enabled dopamine detection at low detection limits in the picomolar range (18.9 pM). The reproducibility of the assay was demonstrated by examining the signal response to interferent compounds coexisting with dopamine in the body (ascorbic acid, uric acid, glucose, epinephrine and serotonin). The nafion membrane attracted dopamine due to its cationic nature of dopamine (pKa = 8.9), while it repelled ascorbic acid because of the presence of anionic sulfonic sites (pKa = 4.04) in its molecular chain. Other analytical parameters, including the reproducibility and repeatability, were also satisfactorily evaluated.

#### 3.4.2. Synovial

Synovial fluid is a viscoelastic solution that functions as a biological lubricant within the cavity of synovial joints. The major constituent of synovial fluid is hyaluronic acid, a mucosubstance which plays an important role in lubricating the articular cartilage. Other chemical components, including lubricin, proteinases and collagenases, are also present in small quantities in synovial fluid. The importance of synovial fluid for clinical diagnosis and prognosis is determined by regulatory cytokines and mediators associated with chronic inflammation [35].

In this line, spiked equine synovial fluid was used to detect cytokines by means of an SPRi immunoassay [26]. The assay consisted of an antibody sandwich cascade with subsequent addition of neutravidin and biotinylated gold nanoparticles as signal amplification. This multiplexing approach allowed simultaneous detection of IL-1β and IL-6 cytokines in synovial fluid with high sensitivity (fg mL^−1^) and low variability in comparison with buffer samples. Although the study of specificity and cross-reactivity with regard to the affinity of antibody pairs is also presented, comparison between equine and human patient samples should be required in order to validate the assay for clinical analysis.

#### 3.4.3. Tears

Tear fluid is a complex biological fluid that contains water, proteins, lipids, glucose, mucins and electrolytes. Tears play an important role in ocular surface lubrication, thus preventing dryness and protecting from external irritants. The majority of the protein content is composed of lysozyme, lactoferrin, lipocalin, secretory IgA, serum albumin and lipophilin [102].

Tears fluid has mainly attracted interest in the field of clinical biomarker detection for the potential of protein detection, since tears are involved in the immune response and inflammatory processes. Therefore, tears are an emerging biofluid for clinical diagnosis of specific disease states and systemic chronic conditions.

An LSPR platform making use of ionic poly(N-isopropylacrylamide-co-methacrylic acid) (PNM) hydrogels synthesized on the surface of silica gold nanoshells (AuNSs) was used to detect lysozyme and lactoferrin, two protein biomarkers that present significantly lower concentrations in patients with chronic dry eye [103]. The electrostatic attraction between the negatively charged hydrogel and the positive charges of the high-isoelectric-point proteins generated concentration-dependent LSPR shifts due to the binding of lysozyme and lactoferrin to the AuNS@PNM-modified surface. LPSR measurements were performed using diluted pooled human tears diluted in either HBS or PBS (1:10). After achieving baseline lysozyme and lactoferrin concentrations, optimized assay conditions were selected, taking into account dilution factor, buffer and AuNS@PNM concentration. Since the sensing scheme only relies on shifts in the LSPR wavelength of AuNS@PNM, no sensitivity values are presented. Moreover, differential sensing of lactoferrin and lysozyme should be implemented by using hydrogels with additional ionic monomers.

## 4. Conclusions

The detection of biomarkers in biofluids is a pressing challenge that is prompting the integration of plasmonic biosensors in clinical settings. Although the number of plasmonic applications focused on measuring clinical biomarkers in blood, urine and saliva samples has substantially increased in the last decade, the search for a universal antifouling strategy is still ongoing.

In many cases, surface biofouling is only prevented by selecting the appropriate sample dilution ratio, particularly in biological media with lower interference, such as urine. However, the design of ultralow-fouling surfaces is the preferred approach to minimize nonspecific adsorption of biomolecules, cells or bacteria. In this sense, plasmonic biosensors have exploited a wide variety of antifouling materials, including PEG/OEG, zwitterionic polymers or polysaccharide-based hydrogels. Nevertheless, the heterogeneity of complex biological matrices limits the extension of a general antifouling method, and the majority of plasmonic applications rely on the specific characteristics of each sample.

As a consequence of the presently required specificity in antifouling methods, the amount and nature of the foulants present in biofluids determine the extent to which the disruption of the analytical signal may interfere the sensitivity and specificity of the assay. From this perspective, the improvement of the analytical performance may extend the enhancement of sensing schemes to the fabrication of stable antifouling materials that strengthen the resistance of the immobilized receptors. Therefore, more efforts are needed to explore not only the antifouling properties of new materials but also the antifouling effect regarding the composition of the media and the bioactivity of the sensing surface after immobilization.

## Figures and Tables

**Figure 1 biosensors-10-00063-f001:**
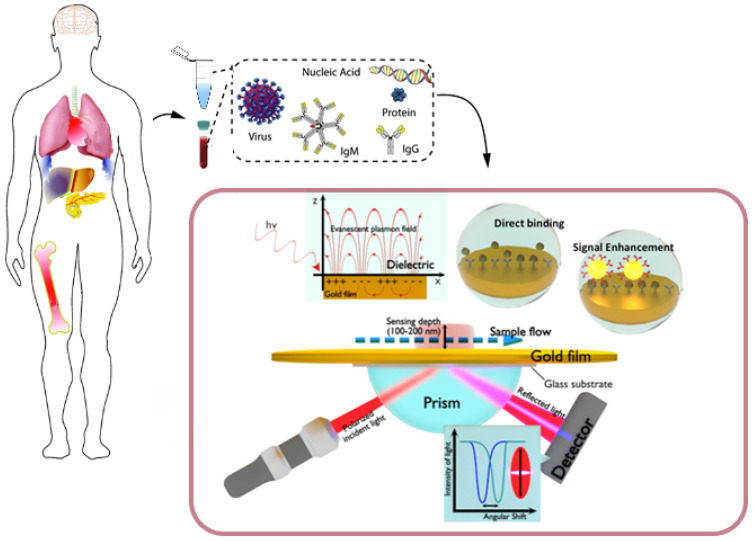
Clinical (plasma/serum/saliva/urine) biomarkers, including virus particles, nucleic acids, proteins and antibodies, that can be monitored by surface plasmon resonance (SPR)-based biosensors. The scheme in the box below represents a plasmonic sensing system based on the Kretschmann configuration. The incident light passes through a glass prism before being reflected by the sensing surface and captured by a detector. The refractive index at the interface and the surface plasmon wave frequency will change upon the binding of analytes. The amount of bound molecules can be measured in real time at a fixed incident angle or by tracking angle–SPR-resolved responses. The inset in the background depicts the generation of a surface plasmon wave that will propagate along the conductor–dielectric interface upon interaction with an incident plane-polarized light. Adapted with permission from Fehran et al. [10] (Copyright © (2018) Elsevier), Rapisuwon et al. [5] (under the terms and conditions of the Creative Commons CC BY License) and Chen et al. [15] (Copyright © (2019) Elsevier).

**Figure 2 biosensors-10-00063-f002:**
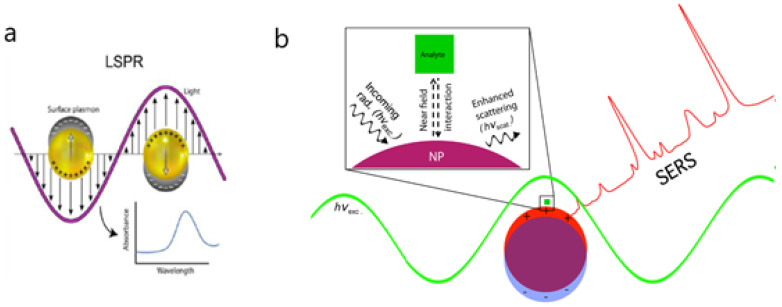
(**a**) Representation of the localized surface plasmon on nanoparticles and absorbance spectra obtained for binding events on nanoparticles. (**b**) Schematic diagram depicting the electromagnetic enhancement of surface-enhanced Raman scattering (SERS). Incoming radiation of resonant wavelength (*h*ν_exc_) interacts with the nanoparticle, exciting a localized surface plasmon resonance (LSPR). The near-field interaction between the Raman scatterer (i.e., analyte) and the plasmonic nanostructure increases the intensity of the scattered light (*h*ν_scat_). Adapted with permission from Masson et al. [24] (Copyright © (2020) Royal Society of Chemistry) and Strobbia et al. [17] (under the terms and conditions of the Creative Commons CC BY License).

**Figure 3 biosensors-10-00063-f003:**
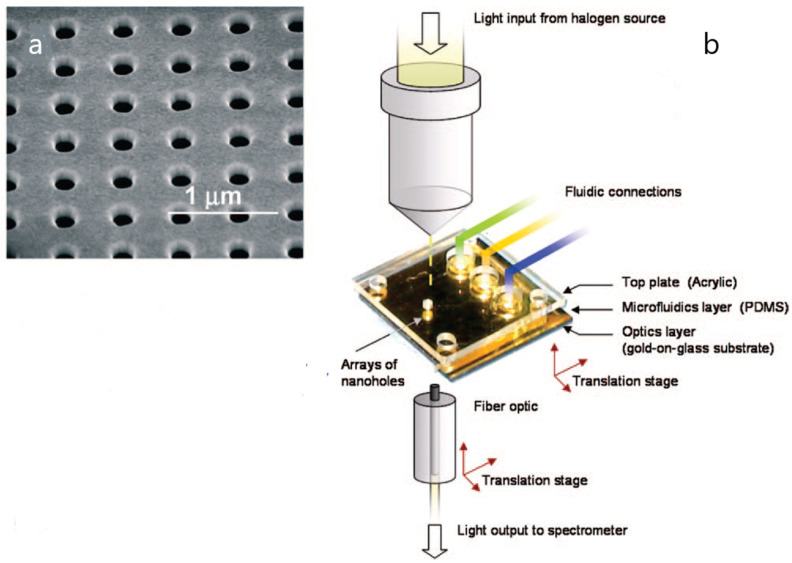
(**a**) Scanning electron microscopy (SEM) image of an array of nanoholes in a gold film; (**b**) experimental setup of a plasmonic-based nanohole array biosensor. Adapted with permission from Gordon et al. [29] (Copyright © (2008) American Chemical Society).

**Figure 4 biosensors-10-00063-f004:**
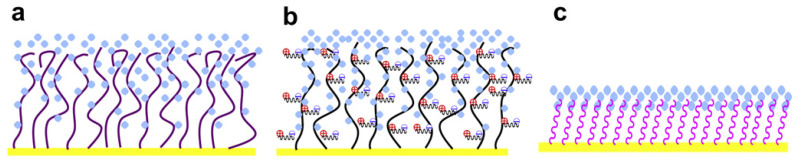
Illustration of chain hydration and chain flexibility of (**a**) hydrophilic polymers, (**b**) zwitterionic polymers and (**c**) SAMs, which have different attributes of surface resistance and nonspecific protein adsorption. Reproduced with permission from Chen et al. [42] (Copyright © (2010) Elsevier).

**Figure 5 biosensors-10-00063-f005:**
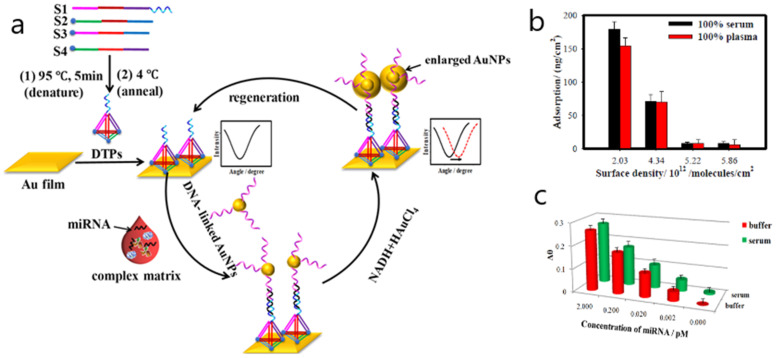
(**a**) Schematic illustration of low-fouling SPR biosensor for high-sensitivity detection of miRNA in complex matrix based on DNA tetrahedron. (**b**) Effect of surface density of DNA tetrahedron probes (DTPs) on nonspecific adsorption (the amount of protein adsorption decreased with the increase of surface density until the surface density of 5.22 × 10^12^ molecules per cm^2^ in 100% serum and 100% plasma). (**c**) Detection of let-7a in 100% human serum (green histogram) and buffer (red histogram). Adapted with permission from Nie et al. [40] (Copyright © (2018) American Chemical Society).

**Figure 6 biosensors-10-00063-f006:**
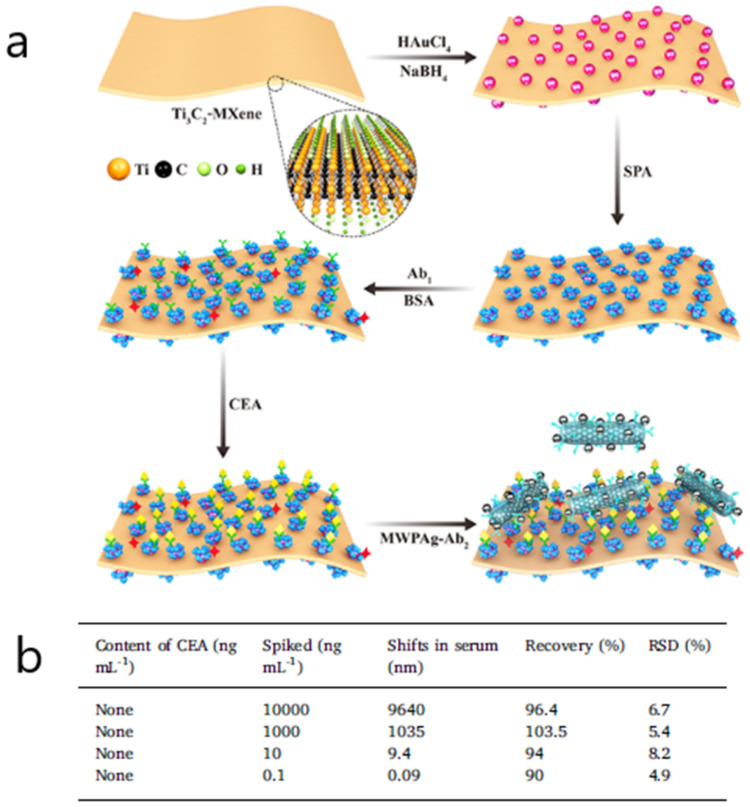
(**a**) Schematic representation showing the detection procedure of the SPR biosensor consisting of a Ti_3_C_2_-MXene-based sensing platform and a multiwalled carbon nanotube (MWCNT)–polydopamine (PDA)–Ag nanoparticle (AgNP) signal enhancer (Ab: antibody; APA: staphylococcal protein A). (**b**) Analytical results of carcinoembryonic antigen (CEA) in human serum. Adapted with permission from Wu et al. [74] (Copyright © (2019) Elsevier).

**Figure 7 biosensors-10-00063-f007:**
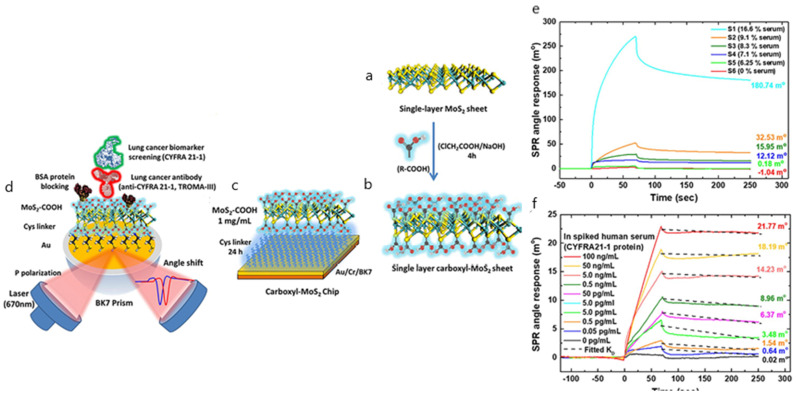
(**a**) Scheme of the synthesis route of single-layer MoS_2_; (**b**) single-layer carboxyl-MoS_2_ nanocomposites; (**c**) the carboxyl-MoS_2_-based SPR chip; (**d**) the carboxyl-MoS_2_-based SPR sensing mechanism to detect the lung cancer biomarker CYFRA21-1. SPR sensorgrams were analyzed for the CYFRA21-1 protein in spiked human serum samples using the carboxyl-MoS_2_-based chip, showing (**e**) different serum concentrations that were used to assess interference analysis during the test and (**f**) comparisons of different serum ratios in SPR biosensing for interference analysis. Adapted with permission from Chiu et al. [78] (under the terms and conditions of the Creative Commons CC BY License).

**Figure 8 biosensors-10-00063-f008:**
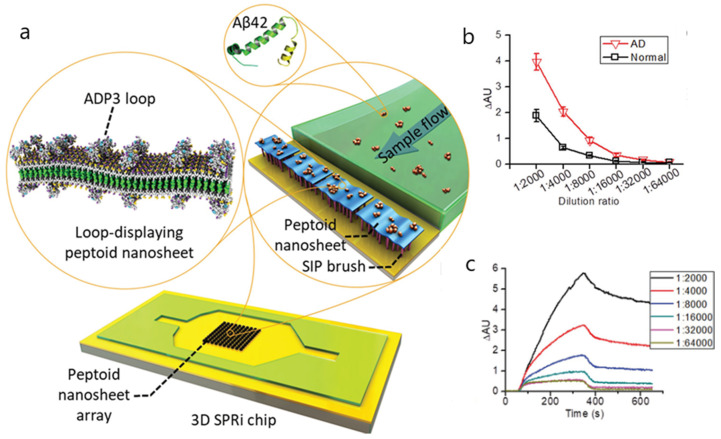
(**a**) Schematic illustration of the detection of Alzheimer’s disease (AD) serum by loop-displaying peptoid nanosheets in combination with 3D surface plasmon resonance imaging (SPRi) sensor chip. The AD peptoid 3 (ADP3) loop-displaying peptoid nanosheets were immobilized on the 3D sensor chip with carboxylated poly(OEGMA-co-HEMA) brushes fabricated on the gold surface by surface-induced polymerization (SIP). As the serum sample flow passed through the chip, Aβ42 in the serum was captured by the ADP3 loop in the nanosheet, generating SPRi binding signals. (**b**) Evaluation of the sensitivity of the ADP3 loop-displaying peptoid nanosheets to detect AD sera. SPRi binding signals of nanosheets with 100% ADP3 loop to AD and normal sera at different dilution ratios are shown. Error bars represent the standard deviation (n = 14). (**c**) Representative SPRi sensorgram showing the binding of nanosheets with 100% ADP3 loop to AD serum at different dilution ratios. Reproduced with permission from Zhu et al. [81] (Copyright © (2017) WILEY-VCH).

**Figure 9 biosensors-10-00063-f009:**
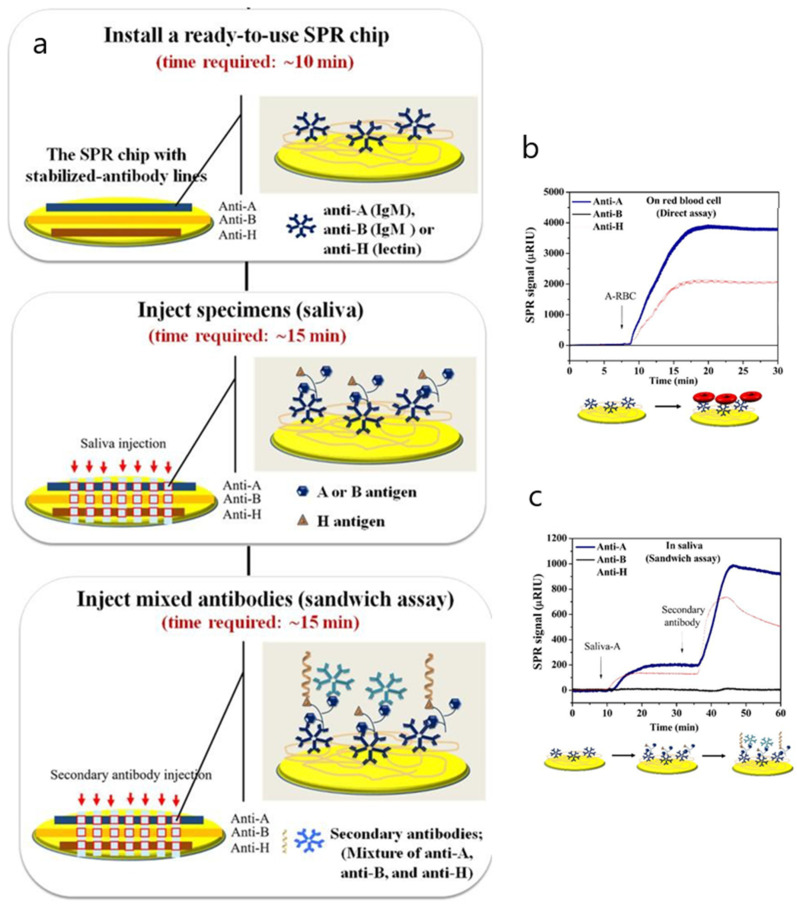
(**a**) SPR procedure for ABH antigen detection in saliva. (**b**,**c**) SPR sensorgrams of ABH antigen detection in red blood cells (direct assay) (**b**) and saliva (sandwich assay) (**c**). Adapted with permission from Peungthum et al. [94] (Copyright (2017) © Royal Society of Chemistry).

**Table 1 biosensors-10-00063-t001:** Key analytical features of clinical diagnostics by plasmonic biosensors classified according to the characteristics of biological samples, detection format (namely instrument configuration or biological receptor) and sensitivity or detection range.

Biofluid Type of Biomarker	Detection Format (Instrument Configuration/Biological Receptor)	Antifouling Strategy/Sample Dilution	Detection Range/LOD	Reference
Blood (serum, plasma)				
Nucleic acids				
MicroRNA	SPR (gold nanoparticles)	DNA tetrahedron probes/ 100% serum, 100% plasma, 9.85 × 10^8^ red cells/mL, 5% whole blood and cell lysate)	0.8 fM (miRNA-429, random miRNA)	[40]
	SPR (near-dispersionless microfluidic system)	Copolymer brushes: zwitterionic carboxybetaine and chains of OEG-SAMs/ Plasma healthy volunteers	0.6 nM (miRNA16)	[59]
	SPRi (orthogonal signal amplification)	CYTOP fluoropolymer/ 10%–40% diluted spiked human serum	3.46–10.5 fM (cancer and healthy 40% diluted serum); 0.56 fM (10% diluted serum) (miRNA15a)	[60]
	SERS (Ag/Au nanospheres-based probes)	Target-catalyzed hairpin assembly/ 10% real human serum samples	0.306 fM/1 fM to 10 nM (miR-133a)	[61]
Circulating tumor DNA	LSPR (gold nanorods)	Spiked healthy patient serum	2 ng mL^−1^	[62]
Circulating tumor cells	SPR (grating coupled fluorescence imaging/antibody imprinting)	BSA 2% blocking/ Mouse blood diluted (1:100)	N/A	[65]
	LSPR (enhanced electrochemistry, gold nanostars, aptamers)	Glassy carbon electrode	5 to 10 cells/mL	[66]
Exosomes	SPRi (AuNPs/antibody)	Incubation in 1% (*w*/*v*) BSA/ Plasma healthy volunteers	10^4^ particles/μL	[68]
	SPRi (array of antibodies)	Carboxylated PEGs/ Plasma (5 healthy volunteers)	N/A (increased the signal response from 5% to 20%)	[69]
Cancer-Related Proteins				
- PSA	SPR (fiber optic, nanopatterned enhancement/antibody)	Diluted serum samples 1:13/ 8 patients’ serum samples	0.1 pg mL^−1^	[71]
	SPR (direct detection/antibody)	Serum samples diluted serially to 1:10, 1:100, 1:1000 and 1:10,000 times (rabbit serum samples)	N/A (protein content in serum samples)	[72]
- CEA	SPR (Biofunctional gold nanoparticles, sandwich assay/antibodies)	Plasma diluted to 30% with PBS/BSA.	N/A	[73]
	SPR (carbon nanotube (MWCNTs)-polydopamine (PDA)-Ag nanoparticles (AgNPs), sandwich assay/polyclonal antibody)	Carbide nanosheets of Ti_3_C_2_-MXene/ 5 cancer patient’s serum samples	0.07 fM	[74]
- Alpha fetoprotein	SPR sandwich assay, direct detection/antibody)	Serum spiked samples	Concentration range 25–400 ng mL^−1^	[75]
	SERS (antibody)	Diluted blood serum (1:250)/ 3 patient samples	0.078 ng mL^−1^ ng/mL (AFP-L3)	[76]
- CA125/MUC16 ovarian tumor biomarker	SPRi (cystamine linker/antibody)	Diluted Serum/ 9 endometriosis samples	2.2–150 U/mL	[77]
- Cytokeratin 19 fragment	SPR (carboxyl-functionalized molybdenum disulfide nanocomposites/antibody)	Different spiked ratios of spiked serum samples (16.6%–0%)	0.05 pg mL^−1^ (6.25%)	[78]
- Folic acid	SPR (graphene-on-metal interfaces/folic acid receptors)	Immersion of folic receptors in mixtures of human serum and bovine serum albumin (BSA) with a volume ratio (1:1).	10–800 fM	[79]
Alzheimer biomarkers				
- Amyloid beta 1–42 (Aβ42)	SPRi (peptoid nanosheet/antibody-mimetic)	Dilution ratios (1:2000 to 1:64,000)	1–10,000 pM	[81]
- Protein Tau	SPR (Sandwich assay/aptamer-antibody)	Alkanethiol mixtures and PEG molecules/ Spiked undiluted plasma	50 fM (Tau-381)	[82]
Inflammation biomarkers				
- Interleukins	SPRi (citrate-stabilized Fe_3_O_4_@Au core/shell nanoparticles (NPs)/sandwich antibody–antigen-antibody)	Iron oxide/gold core/shell nanomaterial/ 10% diluted spiked serum	10 pM–100 nM (IL-6) 8 pM–75 nM (IL-8) [miRNA-21: 50 fM–2 pM miRNA-155: 25 fM–4 pM]	[28]
Infectious agents				
- Hepatitis B	SPR (antigen HBsAg)	Copolymer poly[(N-(2-hydro-xypropyl) methacrylamide)-co-(carboxybetaine methacrylamide)] brush/ Volumetric ratio 1:10, 8 healthy donors	0.0002 to <1 IUmL^−1^	[85]
- Brucella	SPR (4-mercaptobenzoic acid (4-MBA)/DNA probes)	Real serum samples 1:6400 dilution, 10 donors	15.3–54.9 pM (DNA probes of IS711 gene)	[86]
Saliva				
Cortisol	LSPR (gold nanoparticles of different sizes/aptamer)	Centrifuged and filtered real saliva samples	0.01 nM	[91]
Hepatitis B	SPR (amplified fluorescence/hepatitis B surface antigen)	Brushes of poly[(N-(2-hydroxypropyl) methacrylamide)-co-(carboxybetaine methacrylamide)]/ Clinical saliva samples	0.01 to >1 IU/mL (anti-HBs)	[92]
Metalloproteinase-9	SPR (immunosensor/antibody)	Carboxymethyldextran hydrogel/ Filtration and dilution in TRIS buffer, periodontitis samples	8 pg mL^−1^ (0.087 pM)	[93]
ABH antigens	SPRi (multiplex format, sandwich assay anti-A, anti-B and anti-H antibodies)	Carboxymethyldextran, 64 real saliva samples/ Centrifugation, boiling, freezing	N/A precision at 0.06%–4.9% CV.	[94]
Urine				
Glucose	SPR fiber optic (phenylboronic-acid-modified Au nanoparticles)	Spiked urinary samples	2.8 mM	[96]
Human serum albumin	SPR (gold nanoparticles)	Ethylene glycol dimethacrylate and N-methacryloyl-l-leucine/ Spiked urine samples	1.9 pM	[97]
MicroRNA	t-SPR (transmission surface plasmon resonance, capped gold (CG) nanoslit multiplex target microRNA, microfluidic polymethylmethacrylate)	Spiked healthy volunteers samples	30 fM (monitoring probes (I, II, III and R)	[98]
3-Nitrotyrosine	SPR (immunosensor, indirect competitive immunoassay/3NT-OVA conjugates)	Artificial urine, spiked human urine samples	0.12 μg mL^−1^	[99]
Gluten immunogenic peptid (33-mer α2-gliadin)	SPR (indirect competitive assay/gluten immunogenic peptid)	BSA 10 mg mL^−1^ 2 min blocking/ 21 urine samples collected from 2 groups of donors	1.6–4.0 ng mL^−1^	[100]
Cerebrospinal Fluid				
Dopamine	SPR (multiwalled carbon nanotubes/electrostatic interactions)	Polypyrrole polymer imprinting dopamine cation exchange polymer, nafion/ Spiked artificial CSF	18.9 pM	[101]
Synovial Fluid				
Interleukins	SPRi (multiplex format/antibody sandwich cascade)	Spiked equine and human patient samples	2 fM (IL-1β and IL-6)	[26]
Tears				
Lysozyme and lactoferrin	LSPR (silica gold nanoshells/electrostatic attraction)	Poly(N isopropylacrylamide-co-methacrylic acid) (PNM) hydrogels/ Diluted pooled human tears in either HBS or PBS (1/10). Healthy vs. dry eye patients	N/A	[103]

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
