# Peer review of "Low-Fouling Substrates for Plasmonic Sensing of Circulating Biomarkers in Biological Fluids"

_biosensors, 2020, doi:10.3390/bios10060063_

Round 1
Reviewer 1 Report
This review focused on anti-fouling for plasmonic sensing techniques applied in clinical applications. Both anti-fouling surfaces and applications for different biofluids were covered. Comments are as follows.
1: I think the current title maybe misleading. It gives readers the impression that plasmonic techniques were used to achieve low-fouling. If I could suggest, “Low-fouling for plasmonic detection applied in clinical biomarkers in biological fluids” may be better.
2: For some statements in the review, there are no explanations to support them.
For example, the first sentence on page 2. Why plasmonic platform can provide a solid framework?
3:It should be “femto” not “fento” in the unit.
4: There are capital letters in the middle of some sentences. For example, the sentence right before 3.2 section. There is a “No” in the middle.
5: Since this review focused on plasmonic sensing techniques, more information should be included regarding these techniques. For example, the nano-structures support plasmonic phenomena, the excitation frequency, and Raman spectra for SERS sensors should be included.
6: There is a statement in the abstract, “covering all aspects of plasmonic technologies”. I think I cannot agree with this.
7: Some sentences need to be reorganized to make it clearer and more concise. Some sentences are not correct.
8: There is repeated information through the review which makes some content redundant. The review can be reorganized better to make the flow more smoothly.
Reviewer 2 Report
This manuscript presents a review of the advances on antifouling approaches for the plasmonic-based detection of biomarkers in biological media. The manuscript is well presented in an organized manner with subsections that flow well.
The review is comprehensive from an antifouling viewpoint. From a detection technology point of view, it covers well flat surface sensors (SPR, SPRi, SERS) and nanoparticles (LSPR, SERS). However, it leaves out an important approach to SPR-based biosensing that represents a whole area within the field: plasmonic nano-apertures (nanoslits, nanoholes, etc.). In the introduction section, the author mentions and highlights "novel plasmonic sensing methods" which would include the metallic nanostructures which are more modern than established Otto or Kretshmann configurations. Contributiuons using nanoapertures are important and widespread, and include cell studies and pathogen detection (viruses, bacteria). I strongly suggest the author to add these nanostructures that will enhance and will widen the scope and readership of the review significantly. I believe that a subsection (with corresponding figures), adding information to the introduction and inclusion of a couple of figures will suffice.
The samples (i.e. blood, saliva, etc.) section is well covered.
Additional comments:
Pg. 2, second paragraph: an explanation of the differences between the different plasmonic approaches (LSPR, SPR, SERS, etc.) would add a lot of value to the review. An additional figure highlighting these differences will be highly valuable as well.
Pg. 5, subsection 3 title: correct typo "Plamonics"
Round 2
Reviewer 1 Report
The quality of the revised manuscript was improved a lot.
A few mistakes still present.
For example, "Kretschman" should be "Kretschmann".
Reviewer 2 Report
The reviewers' comments and suggestions have been mainly addressed, but I still have concerns about the cited work used to refer to concepts in the introduction section. In pg. 2 (added text, in red), for instance, concepts are not accurately referenced. Here, a natural reference would be one of the 3 most cited articles from Jiri Homola since antifouling has not been introduced yet. Instead, the author cites "Graphene-Gold Metasurface Architectures for Ultrasensitive Plasmonic Biosensing", "Design and mechanisms of antifouling materials for surface plasmon resonance sensors" and "Revisiting the Surface Sensitivity of Nanoplasmonic Biosensors". These references are not as accurate as they should be.
The same thing occurs in pg. 4 ("In the same vein..." paragraph) where several nanoaperture (subwavelength) plasmonic structures are mentioned. The only reference used here is not enough and is a bit more concerning as it is a self-citation. I understand that the author has contributed with previous reviews and done research work in the field, but this reference alone does not fulfill the citation need in this sentence (and section). I strongly encourage the author to employ more accurate references for the presented concepts and previous work. Below, there is a list of (some) suggestions with work on nanoapertures for the author to consider, but should not be limited to. A comprehensive revision of the cited literature is strongly recommended.
- Nature Photonics volume 6, pages709–713(2012)
- https://doi.org/10.1039/C4AN02258K
- DOI: 10.1039/C3LC50107H
- https://doi.org/10.1021/ar800074d
For SERS:
- https://doi.org/10.1039/C0CP01841D
- https://doi.org/10.1021/nl048818w
- https://doi.org/10.1021/nl048818w
Other research-based biosensing articles to consider citing in nanoaperture intro section:
- https://doi.org/10.1039/C3AN36616B
- https://doi.org/10.1021/nl103025u
- 10.1039/C0SC00365D
- https://doi.org/10.1016/j.bios.2018.01.055
It is appreciated that the author considered the recommendation of adding nanoapertures and it does provide a wider coverage and thus more comprehensive review. However, the figure for the working principle is missing (or the suggestion was not considered). I still believe that the addition of a figure with the working principle of plasmonic sensing with nanoapertures will be valuable for this review. It would be straighfowward to add.
